# Durability and Corrosion Resistance Test of Adhesive Joints Using Two Adherence Promoters for the Connection of Aerospace Aluminum Alloys

**DOI:** 10.3390/ma15248733

**Published:** 2022-12-07

**Authors:** Monika Chomiak, Michał Sałaciński, Filip Gołębiowski, Piotr Broda

**Affiliations:** 1Department of Theoretical and Applied Mechanics, Faculty of Mechanical Engineering, Silesian University of Technology, S. Konarskiego 18A Street, 44-100 Gliwice, Poland; 2Air Force Institute of Technology, Airworthiness Division, Księcia Bolesława 6 Street, 01-494 Warsaw, Poland; 3Brenntag Polska Sp. Z o.o., J. Bema 21 Street, 47-224 Kędzierzyn-Koźle, Poland; 4LG Energy Solution Wrocław Sp. Z o.o., LG 1A Street, Biskupice Podgórne, 55-040 Wroclaw, Poland

**Keywords:** adherence promoters, adhesive joints, mechanical properties, aging, aerospace aluminum alloys

## Abstract

This article presents the technology of making an adhesive joint using two primers: Corrosion Inhibiting Primer BR127 (previously used, containing chromium compounds) and, as a potential substitute, Structural Adhesives Primer EW 5000 AS (which does not contain any compounds harmful to the environment). An adhesive film and a sol–gel primer were used to make the joint of two aluminum sheets, and various technologies were used for applying adhesion promoters. The mechanical properties of the prepared samples were tested using two test methods: wedge tests and shear strength tests. In both cases, the samples were aged in laboratory conditions in tap water, and in a climatic chamber (with increased temperature and humidity). The obtained results indicate that the best technology for preparing the joint using each primer is the technology that assumes heating the primer and hardening the adhesive film in one operation. The results of the strength tests indicate that the samples made using the EW 5000 AS primer have higher strength properties under all tested seasoning conditions compared to samples made using the BR 127 primer. It was also confirmed that the presence of moisture and/or water reduces the mechanical strength of the adhesive joints independently of the primer used. The results of the polymer coatings tests to protect the aluminum substrate against corrosion showed that the coatings are only effective for a certain period of time, and, as a result of the NSS test, after 480 h, all the samples were subject to corrosion.

## 1. Introduction

Aluminum bonding technology in aircraft is an area of continuous research and development [1,2,3,4]. Further solutions are being sought to ensure increased strength, stiffness, and durability of adhesive joints [5,6,7]. Technologies for proper surface preparation used for joining attract a lot of attention [8], especially those with adhesion promoters used to repair damaged aircraft areas [9,10,11,12]. At the same time, the increasing awareness of the environmental risk posed by the use of harmful chemical compounds limits their use and forces them to withdraw and replace technologies previously based on them.

In 2009, the US Department of Defense commissioned the development of new solutions that would reduce the use of chromium VI, which is recognized as a carcinogen. Chromium VI, on the other hand, has excellent anti-corrosion properties, and is a component of anti-corrosion coatings and primers widely used in the aviation and automotive industries. Two preparations are available for chromating aluminum: one based on valuable yet toxic chromium (VI) and the other based on non-toxic chromium (III). According to the recommendations US Department of Defense, new non-chromated solutions for corrosion prevention should have the same good parameters as the existing chrome-based ones. Experts from the materials engineering department of the Naval Air Warfare Command: Aircraft Division (NAWCAD) developed a coating with anti-corrosion properties based on an aluminum pigment to replace the currently used chromate solutions [13].

Nowadays, water and solvent-based primers with or without chromates are produced, and have many applications in various industries (e.g., automotive, maritime, equipment, and machine production). Regarding corrosion protection, non-chromated varnishes already meet aviation specification requirements (for some aluminum grades). In order to obtain varnish properties with a chromate pigment, anti-corrosion functions (such as anodic/cathodic protection and long-term effect of chromates) are reproduced in varnishes without this pigment by appropriate chemical additives. The next stage of development is water-based, chrome-free varnishes with a low content of volatile organic compounds (VOCs) and a high content of solid particles, which fully meet the requirements of original equipment manufacturer (OEM) companies with regards to corrosion, application properties, adhesion, and mechanical and chemical resistance [14,15,16,17,18,19,20,21].

Avoiding corrosion is extremely important for airplane safety. If an airplane is damaged due to corrosion, then the stability and safety of any flight are at risk. Moreover, the average service life of an aircraft is between 25 and 30 years. Therefore, the requirements in the aviation industry are more stringent. For example, the varnish must withstand temperatures between −55 and +70 °C, withstand loads during take-off and landing, and protect construction materials against aggressive media, such as hydraulic oils or de-icing agents. Until now, OEMs have applied chromate-containing varnishes to aluminum alloys.

In many cases, only chromate-based primers for corrosion protection were used to protect both aluminum alloys and ferrous metals (then one type of primer is used for everything). Increasing awareness of the risk of chromate primers and recognition by the EU Commission Regulation means that hexavalent non-chromated processes are used more often, both in surface pretreatment and in priming [22,23]. Research is constantly being carried out to optimize the curing parameters of both the primers and the adhesive film, such as the curing temperature and time, allowing the appropriate quality of joints to be obtained, along with the best possible adaptation of the technology to industrial conditions [24,25,26].

The applied primers are used to increase adhesion, protect against corrosion, and cooperate with the sol–gel. The sol–gel method proceeds through a series of chemical processes, including hydrolysis, gelation, drying, and thermal treatment. The process involves the formation of sol through hydrolysis and polymerization reactions and the subsequent attainment of a rigid porous gel. The final product is then obtained by removal of the solvent and residuals from the pores of the gel by aging drying and annealing. The sol–gel process is used in the production of solid materials where the application of layers using this method enables the production of one or multi-component oxide coatings on either glass or metallic substrates. The preparation of thin film with wide surface area via the sol–gel process has unique properties. The process is used in bonding techniques for aircraft, as described in detail in the available literature [27,28,29]. Both tested materials, Corrosion Inhibiting Primer BR 127 [30] and Structural Adhesives Primer EW-5000 AS [31], increase adhesion and protect against corrosion. However, chromate primers, including chromic VI acid, are materials that prevent corrosion due to surface passivation. Moreover, such coatings constitute a group of conversion coatings. These layers are produced as a result of a chemical reaction between chromated metal and chemical compounds of the chromating solution with the use of a practically insoluble salt [32]. Currently, epoxy resin primers are used more and more often it is safe to use. Epoxy primers are mainly used as sealants to provide a perfect, non-porous finish to complement a transparent or translucent topcoat. Epoxy primers produced with the use of solvent technology are used for many different substrates, including (but not limited to) metal, concrete, masonry, wood, and fiberglass.

Additionally, epoxy coatings have water-resistant properties (as well as resistance to other solvents) and, at the same time, inhibit corrosion. Therefore, they are the preferred solution for various applications, including in the construction, automotive, aviation and, above all, marine industries. Studies confirm that they still offer comparably better properties, including shorter hardening time, excellent adhesion and resistance to corrosion, abrasion and solvents. Structural Adhesives Primer with the annotation AS contains corrosion inhibitors [11,12,31]. All alloy additives deteriorate the corrosion resistance of aluminum alloys, the strongest of which are copper and silicon, with the weakest being manganese and magnesium. 

The anti-corrosion properties of aluminum alloys also depend on the microstructure: the lowest resistance is shown by high-strength Al-Cu alloys for plastic working (called duralum) in the structure in which the CuAl_2_ intermetallic phase appears. Aluminum alloys used in technology are divided into foundry and plastic processing. Due to the very favorable mechanical properties and relatively low weight of the elements, they are now used in the aviation industry, despite their lower corrosion resistance. Anodizing and plating are the basic methods of protecting the surface of aluminum alloys against the adverse effects of the environment. In the group of casting alloys, silumines (Al-Si, with a close eutectic composition, without the addition of copper) and Al-Si-Mg alloys, have fairly good corrosion resistance. Copper alloys (duraluminum types), however, are subject to intergranular corrosion. Other alloys, mainly precipitation-hardened in the Mg_2_Al_3_ phase, do not show a tendency to this type of corrosion, since their local anode areas are dispersion precipitations [33].

This paper discusses the issues related to the preparation of the surface before the joining of aluminum alloys used in aviation structures, as well as the permanent single-post adhesive joint production process. The method of making the joint is presented using three different technologies used to apply primers. The preparation of samples using different technologies is associated with the intention to compile the joint technology recommended by the manufacturer (marked in the paper as technology 1), as well as two adaptations consisting of shortening the processing time and removing the requirement of prior primer hardening (designated as technology 2 and technology 3, respectively).

## 2. Experimental Procedure

### 2.1. Materials for Research

The following products were used to make samples in the form of a single-fold adhesive bond of two 1.6- and 3.2-mm thick aluminum alloy 2024-T3 sheets (it is a supersaturated alloy: cold-work hardened and naturally aged, acc. to LAS Aerospace Ltd., Okehampton, UK):Structured adhesive film AF-163-2K (thermosetting epoxy adhesive in the form of a film) by 3M [34];Sol–gel AC-130-2 primer from 3M [35];EW-5000 AS adhesion promoter from 3M [31];BR127 adhesion promoter by Cytec [30].

Table 1 and Table 2 show the chemical composition and heat treatment state of the aluminum alloys 2024 T3 used in the tests, while the basic properties and chemical composition of the tested primers are presented in Table 3.

To achieve a T3 temper, the metal is solution heat-treated, strain-hardened, then naturally aged.

### 2.2. Preparation of the Specimens

The preparation of test specimens began with cutting out the 2024-T3 aluminum alloy sheets with dimensions of 205 mm × 152.4 mm and a thickness of 3.2 mm and was prepared for the production of panels (Figure 1) in accordance with ASTM D3762 [38]. Additionally, sheets with dimensions of 200 mm × 101.6 mm × 1.62 mm were prepared to produce panels (Figure 2) in accordance with ASTM D1002 [39]. According to these procedures, 24 pieces of sheet metal with a thickness of 3.2 mm and 12 forms of sheet metal with a thickness of 1.62 mm were prepared.

The first stage of the surface preparation process was to degrease the forms with acetone to remove any dirt visible on the surface. Next, the forms were processed with P180 sandpaper using the Rotex RO 90 DX eccentric sander by Festool in Wendlingen, Germany. After sanding the top layer, the forms were cleaned with acetone before being heated in an oven at 60 °C for 45 min to evaporate the acetone from the surface. The next stage of process was to apply a layer of aqueous sol–gel solution to the forms, which was applied by hand with a brush. In order to evaporate water from the solution, the pieces were conditioned for 1 h at room temperature. After this time, the tiles were dried in the oven for 30 min at 60 °C. The samples were then removed from the oven and allowed cool in air. Subsequently, adhesion promoters were applied to the prepared surfaces using various technologies (Tech. 1, Tech. 2, and Tech. 3). The described surface preparation method was the same for all technologies. Table 4 summarizes the technologies for applying primers used to prepare glued samples as a one-overlap joint.

Finally, 24 samples from the first form and 21 samples from the second form were prepared with a steel wedge also prepared for each sample (Figure 3).

**Technology 1 (Tech. 1).** Hardening of the primer in accordance with the manufacturer’s recommendations [30,31].

In this technology, the primer is annealed in accordance with the manufacturer’s recommendations after it is applied to the prepared sample surface using the following method:The BR127 primer was annealed at 120 °C for 30 min.The EW-5000 AS primer was baked at 120 °C for 60 min.

Once the samples cooled down after annealing, one layer of the adhesive film was placed on one of the two pieces joined together. Additionally, a non-stick film was applied to the upper surface of the forms from which the panel was prepared in accordance with ASTM D3762 to enable the wedge test at the joint. Next, the protective foil of the adhesive film was removed, and the forms were joined (the forms were joined this way for all technologies). The correct pressure of the form joint forms was ensured during the curing process using a vacuum bag and was carried out in the oven at 120 °C for 90 min. After cooling, the pieces were removed from the vacuum bag and prepared for cutting (Figure 4).

**Technology 2 (Tech. 2).** Curing of primer and adhesive film in one operation.

A primer was applied to the sample surface covered with sol–gel and then heated at 65 °C for 20 min to evaporate the solvents from the primers. After cooling the forms, all subsequent operations were performed using the same method as for Technology 1.

**Technology 3 (Tech. 3).** The process of hardening a primer under room conditions.

Technology 3 assumed no primer annealing at elevated temperatures. After applying the primers to the forms, they were conditioned at room temperature (approximately 24 °C) for 20 h. Then, an adhesive film was applied to the samples. The process of applying and hardening the glue did not differ from the processes discussed for Technologies 1 and 2.

### 2.3. Research Methodology

The test program (Table 5) consisted of two types of mechanical tests to compare the strength properties of the adhesive joint with the use of two primers, the selection of the optimal primer application, and the hardening technology from among the three tested. Two tests for the mechanical properties of the adhesive joint were carried out: the first was the static tension of the adhesive lap joint in accordance with the ASTM D1002 standard, and the second was the gap opening test using a wedge in accordance with the ASTM D3762 standard. Both tests were carried out on samples seasoned in three ways: laboratory conditions (50% RH, 21 °C), immersion in tap water at 23 °C (to minimize the impact of water on the corrosion of steel wedges, the water was changed every 48 h), and finally at high humidity and temperature (95% RH, 50 °C) using a Discovery Climatic Chamber. The aging conditions were selected based on the experience of AFIT specialists (Air Force Institute of Technology in Warsaw, Poland) as well as from previous literature [40,41]. Both indicated that humidity is one of the most aggressive environments to which an epoxy adhesive bond of aluminum aviation alloys may be exposed. To determine the anticorrosive properties of the used primers, corrosion resistance tests were carried out in a Ascott CCP 450ip salt chamber. Both samples were tested without primer and those coated with two previously tested primers (BR 127 and EW 5000 AS).

### 2.4. Wedge Test

At the outset, it should be emphasized that the wedge test performed is a durability test and is not used to measure the mechanical strength of adhesive joints. The result of the test is the length of the crack and the type of damage (cohesive, adhesive, mixed) as well as the place of its occurrence (between the substrate and the primer or between the primer and the adhesive). During the wedge test, the crack propagation progress after driving the wedge per unit of time was measured. For each technology, 48 samples were cut from 2 panels for each primer. To stress the samples, a steel wedge was mechanically inserted instead of anti-adhesive tape (Figure 5). After 3 h, the crack length was measured on each sample with a caliper with an accuracy of 0.01 mm. The place of the end of the crack was marked by cutting the edge with a sharp knife (the determination of the end of the crack area was carried out based on observation with a Delta Optical Smart 5 MP Pro digital microscope). The cut’s distance from the sample’s beginning was then measured with a caliper (the length of the area covered with the release tape was subtracted from the result). Another crack measurement was carried out after 16 h, with subsequent measurements repeated every 24 h until 256 h (excluding holidays and weekends), at which point the cracking stopped propagating, and the test was marked complete. The area where the test was completed was marked with a red line.

After all the measurements were taken, the samples were mechanically separated, and the area of the wedge-induced fracture was visually inspected. At the breakthrough of the samples, the percentage share of the type of cohesive and adhesive crack was assessed. In the case of an adhesive crack, it was determined between which layers the crack occurred (aluminum substrate—primer or primer—adhesive adhesive).

### 2.5. Standard Test Method for Apparent Shear Strength of Single-Lap-Joint Adhesively Bonded Metal Specimens by Tension Loading

In the next step, the tensile shear strength of the joints was compared for each technology (and for both primers) at a specific load increment, allowing for the predictable strength of the joint to be determined, as well as the nature of failure. For this purpose, the strength test of single-lap joints was carried out in accordance with the ASTM D1002 nomination. For each technology, there were 42 samples cut from one panel for each primer. For each joining technology and both primers, the samples were seasoned before the test in the same way as the wedge test. The samples immersed in tap water were seasoned for 570 h, then a batch of samples was seasoned in a climatic chamber under constant conditions for 460 h. In the case of samples seasoned in water, the test was performed on the sample using a ZWICK Z020 (manufacturing in Ulm, Germany) calibrated testing machine using a speed of 1.3 mm/min immediately after it was taken out of the water. After testing, the samples were then visually inspected.

### 2.6. Corrosion Resistance Tests

Corrosion tests were carried out to assess the corrosion resistance of samples with and without permanent anti-corrosion protection. Tests were conducted in an ASCOTT S450ipACC46 (Ascott Analytical Equipment Limited, Staffordshire, UK) salt chamber and sprayed with brine according to the PN-ISO 9227:2017 standard [42]. The great advantage of this type of research is its international standardization and its repeatability, especially in the case of samples prepared and constantly tested in the same laboratory conditions.

Test specimens with dimensions 70 mm × 100 mm were cut from sheets with a thickness of approximately 1.05 mm from the aluminum alloy 2024-T3. Four samples of the material without protection and four samples with BR-127 and AS 5000 coatings were prepared. The tests were carried out in sprayed brine for the corrosivity class C5, which reflects the working conditions in an aggressive environment with high humidity. The environment was determined based on the time the sample could withstand the environment in a neutral salt spray chamber (NSS). The samples were placed in a chamber continuously sprayed with salt for 480 h. Observations were made after 72, 120, 168, 264, 360, 456, and 480 h to capture the first signs of corrosion. The samples were coated on only one side, and no damage was noticed at the edges of the samples.

The exact test conditions, i.e., temperature, humidity, salt spray collection rate, concentration, pH of the sprayed brine, and spraying method, were selected in accordance with the standard [42].

## 3. Research Results

The tensile shear test and the wedge test allowed for the comparison of the mechanical properties of samples prepared with the three different technologies using the two different primers.

### 3.1. Wedge Test Results

The results obtained in the wedge test are summarized in Table 6 and illustrated in Figure 6, Figure 7 and Figure 8, which show the relationship between the crack growth and the aging time under seasoning conditions for each adhesive joint preparation technology, respectively The number of samples tested in each prepared technology and the standard deviations of the average values obtained are given in the research program (Table 5) and in the graphs (Figure 6, Figure 7 and Figure 8).

### 3.2. Results of Apparent Shear Strength of Single-Lap-Joint Adhesively Bonded Metal Specimens by Tension Loading

To illustrate the results obtained in the shear strength tests, the average values of the connection strength are illustrated in Figure 9 and a single graph of the force–displacement curve is shown in Figure 10, Figure 11, Figure 12 and Figure 13.

### 3.3. Corrosion Test Results

At the end of the test cycle (after 480 h of exposure), damage in the form of stains and blisters under the coating was observed on the surface of the samples for both primers. On the other hand, the samples not protected with the coating showed the greatest susceptibility to corrosion, manifested in particular by a rusty bloom spreading from the edge of the sample after 168 h.

Cyclic observations of the changes taking place on the surface of the samples showed that after 72 and 120 h exposure to saline, no significant changes were observed on the samples. The first signs of corrosion appeared only after 168 h. However, while this was true for samples without coating and with BR 127 primer, no changes were noticed on the surface of the samples with EW-5000 AS primer. However, after 264 h (Figure 14), signs of corrosion were visible on all tested samples. After the test was stopped, the samples were air-dried for 1 h before being washed with running brine water and dried with compressed air. Finally, photos of the samples were taken (Figure 15).

## 4. Analysis of Research Results

### 4.1. Analysis of the Research Results of the Wedge Test

In all the analyzed conditions of aging of single-lap joints made with the use of two different primers, the second technology turns out to be the best connection technology, which assumes heating the primer and glue in one operation, which shortens the time of joint preparation, which is extremely important from the point of view of repairing damage to aircraft components. Similarly, the applied technology described with number three as an alternative to the time-consuming technology of primer manufacturers seems promising; the prepared joints showed good adhesive properties, especially in laboratory conditions with a chromium-free primer.

The lowest crack propagation in the wedge test under laboratory conditions was obtained for samples produced using technology 2, which uses the EW-5000 AS primer. The smallest area of an adhesive crack also characterized these samples. Slightly higher crack growth was obtained for samples produced using technology 3, which used the same primer. However, these samples showed a larger area of destruction of the adhesive joint. The highest results were obtained for technology 1. One of the samples clearly showed a worse result, which was dominated by the adhesive nature of the failure. For samples prepared using the BR 127 primer, there are no clear differences in the crack size between the samples produced in technology 2 and technology 3. However, the samples produced in technology 1 showed the highest crack propagation, for which the largest area of an adhesive fracture was also obtained. The results obtained for the second panel are similar and those presented in the literature [43]. 

The crack growth is not noticeable after a short time for samples aged in laboratory conditions, regardless of the sample preparation technology.

The samples seasoned in water with the EW-5000 AS primer were characterized by a lower crack propagation. The best results were obtained for samples made with technology 3, where both the smallest crack propagation occurred and the most favorable damage pattern was obtained (the largest area of the cohesive character of the crack). Slightly higher crack growth was obtained for technology 2, and the samples produced in technology 1 showed the highest crack propagation. For samples made with BR 127 primer seasoned in tap water, the best properties were shown in samples prepared in technology 2, i.e., the lowest crack propagation. The samples made using technology 1 showed the highest crack growth and the largest area of the adhesive character of the crack.

Among the samples aged in a climate chamber, for both primers, the lowest crack propagation was characteristic for the samples produced in technology 2. The samples produced using the EW-5000 AS primer showed higher properties and a smaller area of adhesive character of the crack. The highest crack growth was obtained for samples made with technology 1 using both primers. 

### 4.2. Analysis of the Research Results of Apparent Shear Strength of Single-Lap-Joint Adhesively Bonded Metal Specimens by Tension Loading

In the plots of the results obtained in the shear strength test, it was noted that the increase in elongation is equal to the increase in force. The best strength properties of the joint were obtained for the EW 5000 AS primer using the third technology in all tested conditions. On the other hand, the BR 127 primer worked only in laboratory conditions in aggressive environments, regardless of the technology, the properties were lower. It should also be noted that despite the good durability of the joint made with the EW 5000 AS primer, the shear strength of the adhesive joint is the lowest in the second technology of glued surface preparation.

The lowest shear strength values were obtained for samples conditioned in laboratory conditions, prepared using the EW-5000 AS primer for joints produced using technology 2. The highest strength was obtained when testing samples produced using the same primer using technology 3. Slightly lower results were obtained for the second technology, which involved hardening the primer following the manufacturer’s recommendations, i.e., technology 1. In the case of the BR 127 primer, no clear influence of the surface preparation technology on the strength was noticed since there is no significant difference between the obtained results. In the case of samples produced using technologies 1 and 3, higher results were obtained for joints produced with the use of the EW-5000 AS primer compared with the BR 127.

The highest shear strength results of samples prepared with the use of EW-5000 AS primer, seasoned in water, were obtained for joints produced using technology 3 (as was the case for the reference samples). Slightly lower results were obtained for technology 1 with the use of the same primer and for technology 1 and 2 with the use of the BR 127 primer. The lowest mean value of the shear strength was obtained for the joint prepared using technology 2 with the EW-5000 AS primer.

For samples seasoned in high temperature and humidity in the climatic chamber, the highest average shear strength was obtained, similar to the previous seasoning conditions, for samples made with technologies 1 and 3 using EW-5000 AS primer. Again, the weakest connection turned out to be a sample made using the same primer in technology2. There are also visible differences between the preparation technologies using the BR 127 primer. The highest results were obtained for technology 1, with slightly lower results obtained for technology 2. Technology 3, with the use of BR 127, for samples seasoned in a climatic chamber, turns out to be the weakest.

The lowest shear strength results and the visible incorrect coverage of the glued surface by the adhesive film overlap indicate the rejection of the results of the samples made with technology 1 using the EW-5000 AS primer. Therefore, the process of producing samples for testing the shear strength in this technology should be repeated.

The results indicate that the shear strength values of the joints differ significantly when both primers are used, with the EW 5000 AS primer showing better properties for the first and third surface preparation technologies. On the other hand, the BR 127 primer is more stable, which is visible in the table (Figure 9), and only aggressive tropical conditions significantly reduce the shear strength value by about 20% in relation to laboratory conditions, regardless of the surface preparation technology.

### 4.3. Analysis of the Research Results in the Corrosion Test

The results of corrosion tests show that one-sided surface protection with a primer does not protect against corrosion on the aluminum substrate due to the aggressive environment. The accelerated aging tests were conducted solely to compare the anti-corrosion properties of two different primers.

Even though the primer increases the adhesive joint’s strength, the soil layer does not sufficiently protect the metal against the aggressive environment. Therefore, the metal surface and the joint should be protected so that no moisture gets in. In repair technologies using composites, developed at ITWL (among others) [44,45], this problem has been solved using appropriate sealants to seal the composite surface and the edge of the patch, where an adhesive film is used as the last layer of the composite patch (from the outside) to seal the pores in the composite. On the other hand, the patch edges are secured with an epoxy or polyurethane sealant of hardness 80–90 Shora. 

In addition, the whole thing is covered with a tight paint that protects against moisture penetration. Furthermore, elements repaired via gluing are subject to supervised operation, which consists of an increased number of checks in relation to the undamaged structure in terms of sticking the patch and, for corrosion defects, under the patch. The adhesion condition is checked by thermography, acoustic impedance, or laser shearography. In contrast, the state of corrosion defects under the patch can be checked by the eddy current method using the MOI system [46,47].

Modern solutions propose using nanotechnology-based varnishes as substitutes for varnishes containing chromium VI since the properties of chromate-free varnishes can be improved. Thanks to nanoparticles, the packing density of the paint particles (and thus its strength) can be improved. However, nanomaterials do not constitute widespread use of such coatings due to costs. In the foreground is the selection of the appropriate binder and the composition of the corrosion inhibitor kit. New problems constantly arise for various areas on the plane, new problems constantly arise, the solutions of which are sought in developing newer chromate-free primer formulas in accordance with the REACH directive products without VOC, or at least very low VOC content [48].

## 5. Conclusions

This study aimed to compare two anti-corrosion primers with different chemical compositions used in joining aerospace aluminum alloys in terms of mechanical properties and the level of anti-corrosion protection.

The results obtained in this work have led to the following conclusions:The wedge test showed that the second surface preparation technology used, which assumes heating and hardening of the adhesive in one operation, provides the best bond durability. In addition, in the case of the BR 127 primer, the conclusion is confirmed by the results of shear strength tests.The results of the tensile shear strength of samples made using the EW-5000 AS primer show that these samples have higher strength properties in each tested seasoning condition compared to samples produced using the BR 127 primer. Therefore, it can be stated that the EW-5000AS primer can be a replacement for the BR 127 primer.The shear strength value of the samples seasoning in water or tropical conditions decreased by about 15% for both primers compared to laboratory conditions. The least aggressive environment for seasoning was laboratory conditions, which constituted a specific reference necessary for properly comparing the obtained test results.The results of the corrosion resistance tests of the undercoats protecting the substrate showed, in the case of the tested samples and the test environment used, increased corrosion susceptibility after 264 h of exposure to brine spray.The expected benefits for aviation include using substances without harmful chemical components for bonding, i.e., chromium VI at the appropriate level of mechanical properties and reliability of glued aircraft structures, as well as providing additional guidance for users in the aviation industry.

## Figures and Tables

**Figure 1 materials-15-08733-f001:**
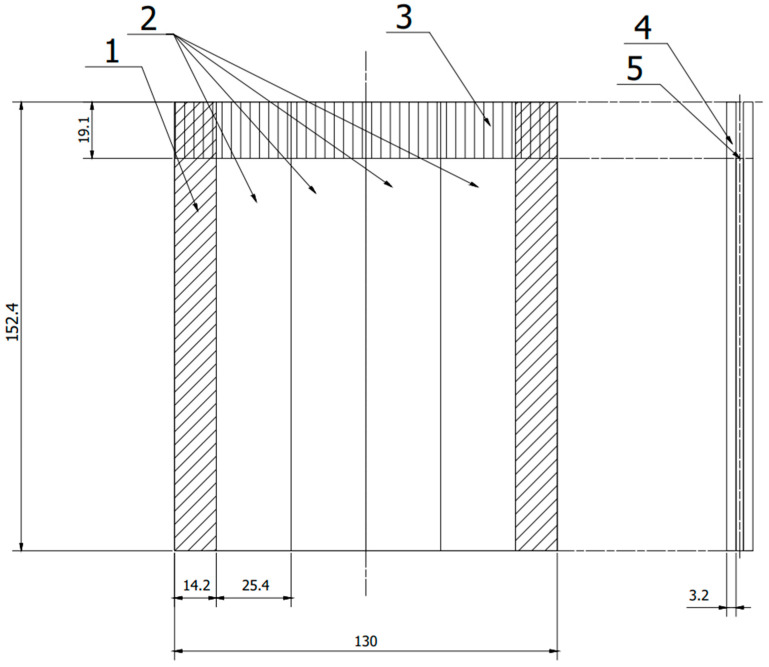
Scheme panel consisting of two plates joined together, from which samples were cut according to ASTM D 3762. 1—allowance rejected during cutting, 2—samples, 3—anti-adhesive tape, 4—sheet, 5—adhesive film. (All dimensions in the drawing are in [mm]).

**Figure 2 materials-15-08733-f002:**
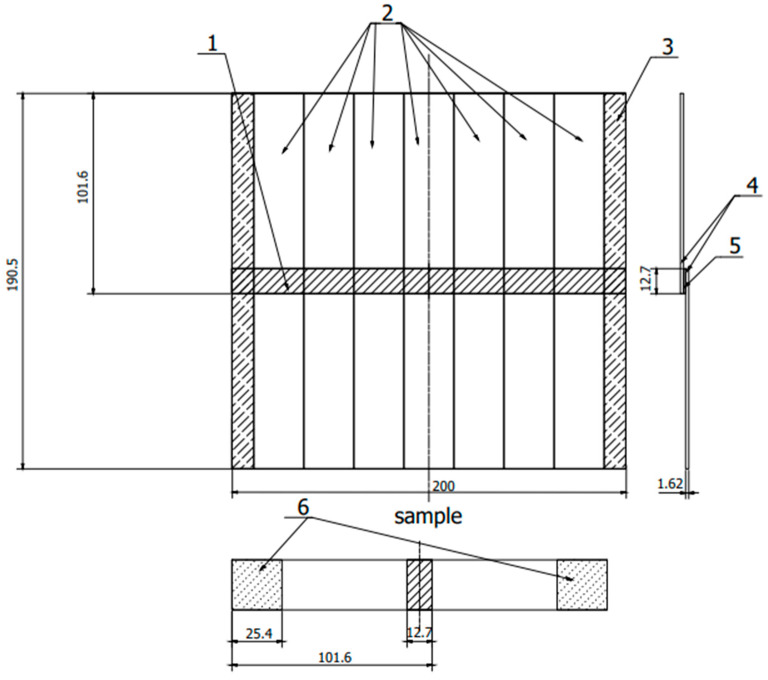
Sample scheme according to ASTM D 1002. 1—adhesive connection, 2—samples, 3—cutting allowance, 4—sheets, 5—adhesive connection, 6—fixing place in the holders of the testing machine. (All dimensions in the drawing are in [mm]).

**Figure 3 materials-15-08733-f003:**
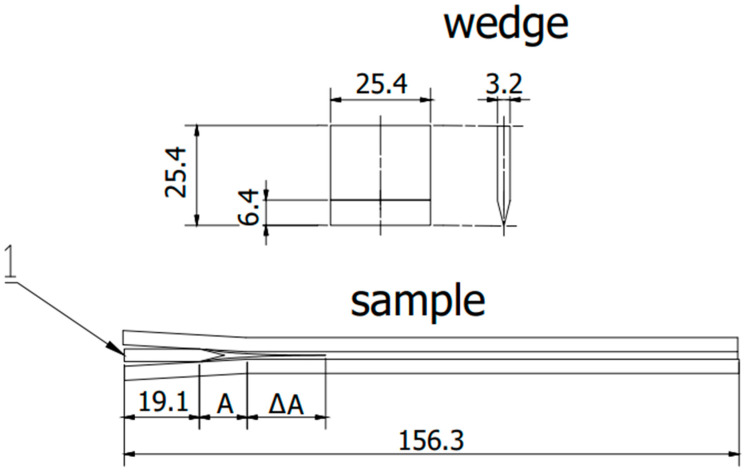
Scheme sample in accordance with ASTM D 3762. 1—wedge, A—initial flow area, ΔA—crack propagation after exposure. (All dimensions in the drawing are in [mm]).

**Figure 4 materials-15-08733-f004:**
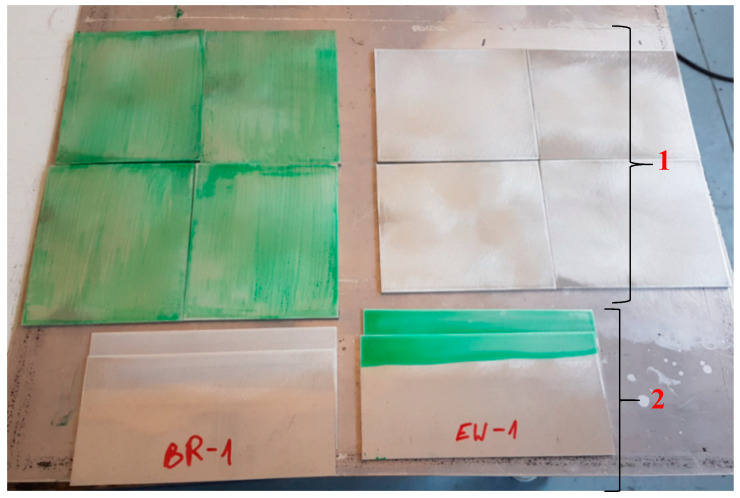
Formats prepared for cutting samples. 1—Sheets prepared to manufacture panels according to the ASTM D3762 standard. 2—Sheets prepared to manufacture panels according to ASTM D1002 standard.

**Figure 5 materials-15-08733-f005:**
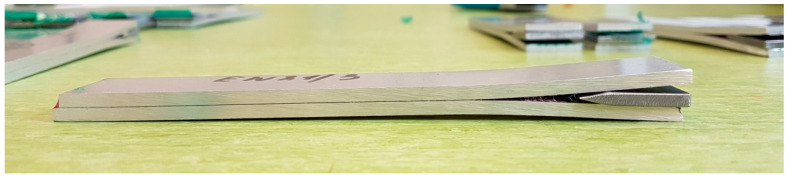
View of the sample during the test.

**Figure 6 materials-15-08733-f006:**
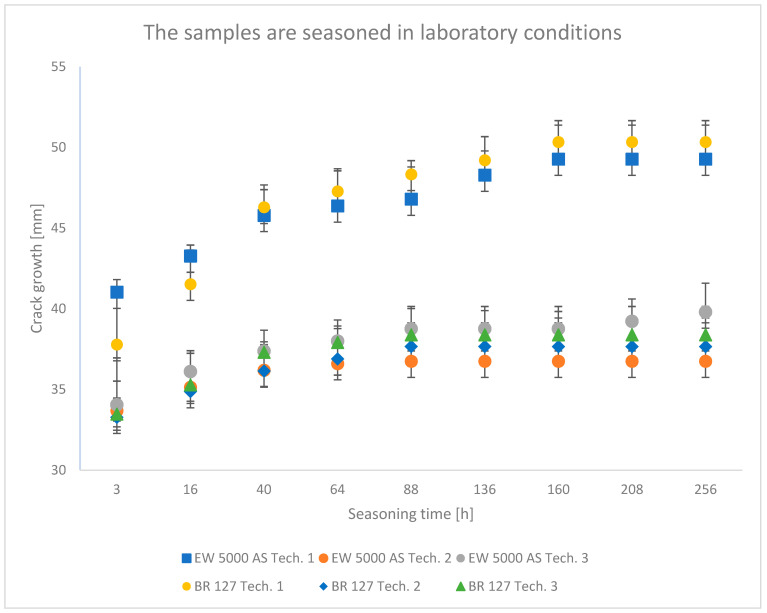
Average crack growth for samples depending on the aging time in laboratory conditions for each type of sample preparation.

**Figure 7 materials-15-08733-f007:**
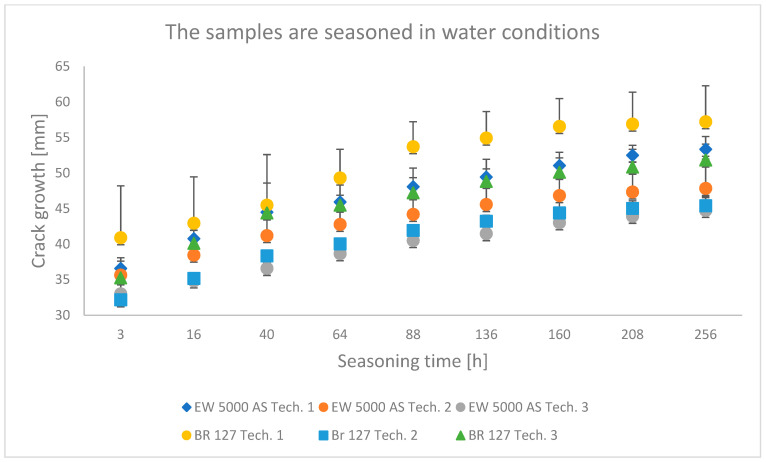
Average crack growth for samples depending on the seasoning time in tap water for each type of sample preparation.

**Figure 8 materials-15-08733-f008:**
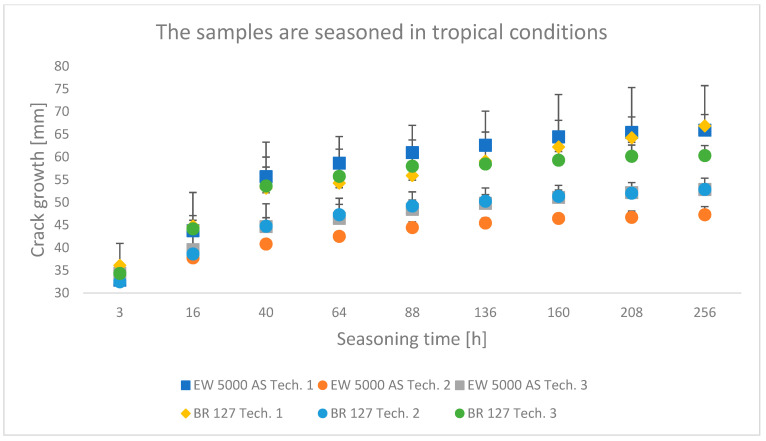
Average crack growth for samples depending on the seasoning time in tropical conditions for each type of sample preparation.

**Figure 9 materials-15-08733-f009:**
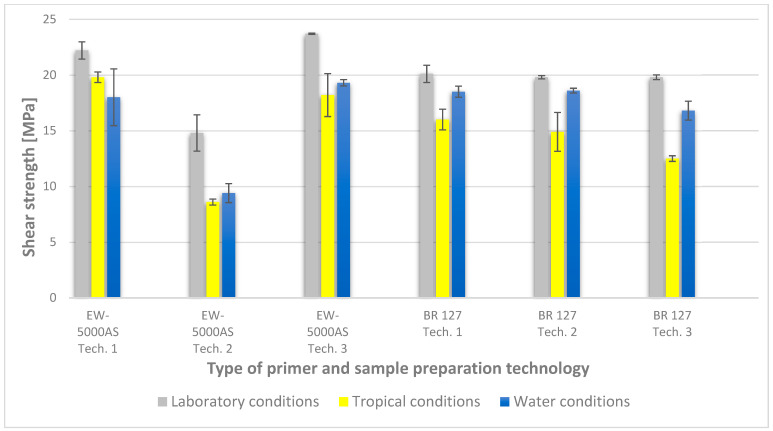
Summary of average shear strength values of samples made with different technologies, depending on the sample seasoning method.

**Figure 10 materials-15-08733-f010:**
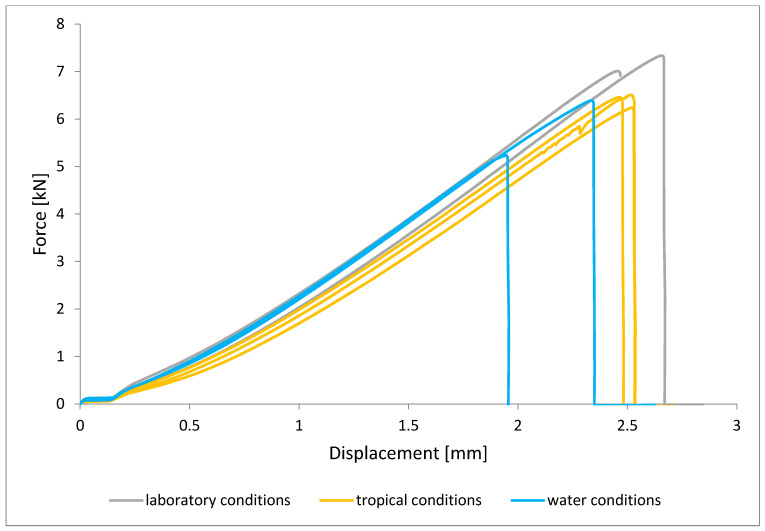
Shear force–displacement curve for single lap joint with EW 5000 AS primer in first technology.

**Figure 11 materials-15-08733-f011:**
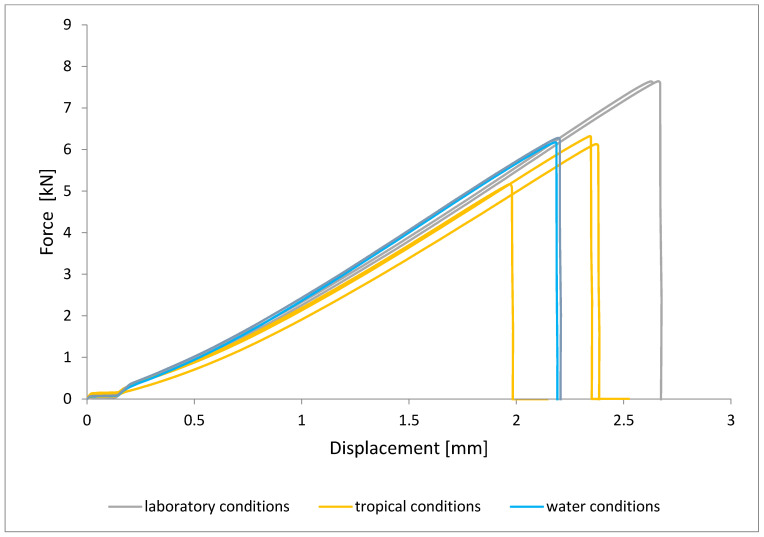
Shear force–displacement curve for single lap joint with EW 5000 AS primer in third technology.

**Figure 12 materials-15-08733-f012:**
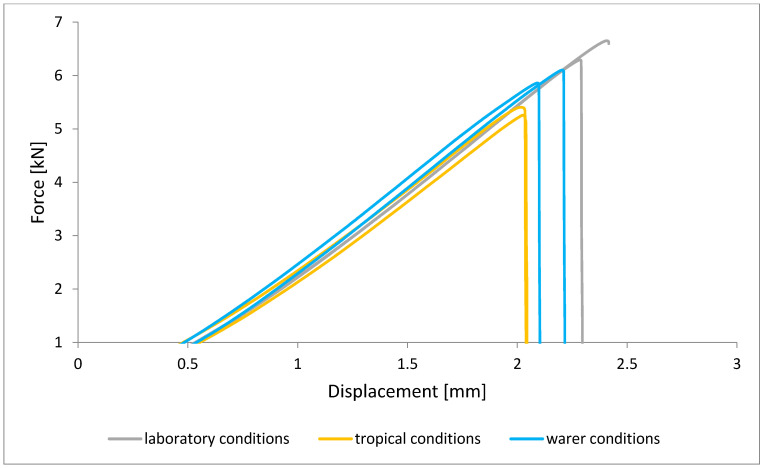
Shear force–displacement curve for single lap joint with BR 127 primer in first technology.

**Figure 13 materials-15-08733-f013:**
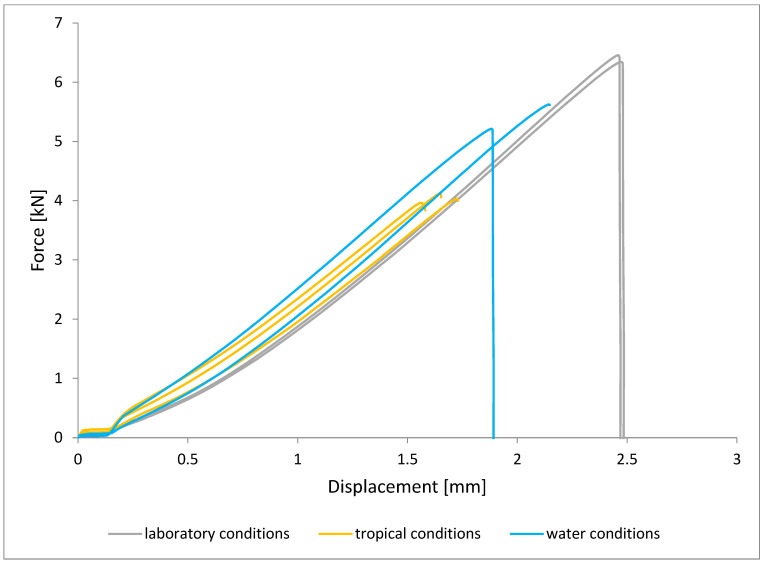
Shear force–displacement curve for single lap joint with BR 127 primer in third technology.

**Figure 14 materials-15-08733-f014:**
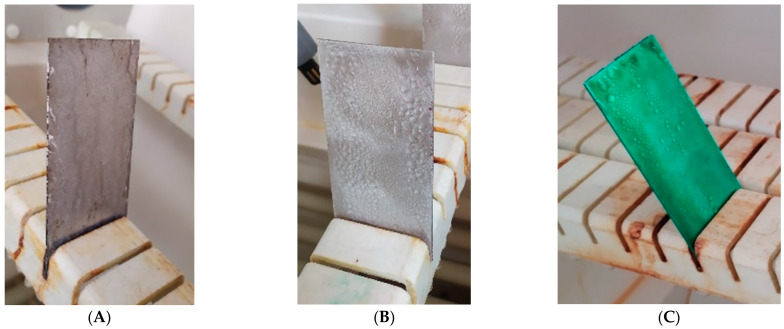
Samples after 264 h of exposure to salt spray. (**A**) Sample without coating. (**B**) Sample with corrosion inhibiting primer BR 127. (**C**) Sample with structural adhesive primer EW-5000 AS.

**Figure 15 materials-15-08733-f015:**
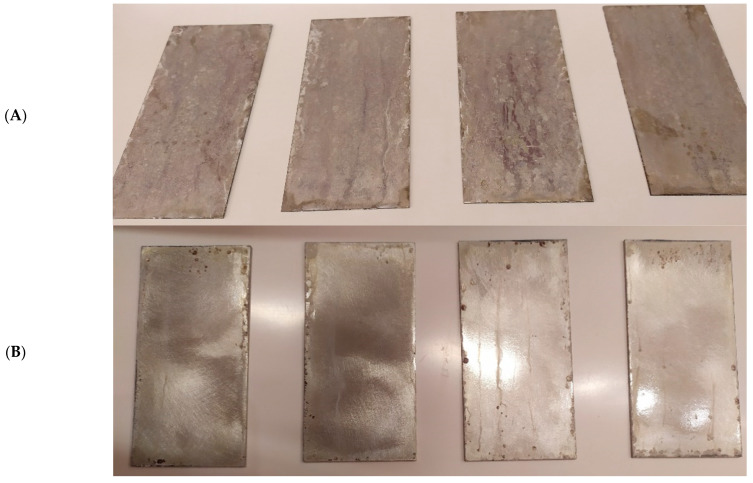
Samples after 480 h of exposure to salt spray. (**A**) Sample without coating. (**B**) Sample with corrosion inhibiting primer BR 127. (**C**) Sample with structural adhesive primer EW-5000 AS.

**Table 1 materials-15-08733-t001:** The chemical composition of aluminum alloy 2024—T3 was used for testing [36].

Component	wt. %	Component	wt. %
Al	90.7–94.7	Cr	Max. 0.1
Cu	3.8–4.9	Fe	Max. 0.5
Mg	1.2–1.8	Mn	0.3–0.9
Si	Max. 0.5	Ti	Max. 0.15
Zn	Max. 0.25	Other, each	Max. 0.05
Other, total	Max. 0.15		

**Table 2 materials-15-08733-t002:** Basic mechanical and physical properties of the aluminum alloy 2024—T3 [37].

Mechanical and Physical Property	Value
Yield Strength, [MPa]	345
Tensile Strength, [MPa]	483
Modulus of Elasticity, [GPa]	73.1
Elongation, [%]	15
Hardness, HB	123
Thermal Conductivity, [W/mK]	121
Density, [g/cm^3^]	2.78
Melting Point, [°F]	935–1180

**Table 3 materials-15-08733-t003:** Basic physical and chemical properties of the tested primers [30,31].

	BR 127		EW-5000 AS	
	Composition	% by wt.	Composition	% by wt.
Information on Ingredients	◆ 2-Butanone (Methyl Ethyl Ketone)	60–75	◆ Water	50–70
	◆ 2-Ethoxyethanol	10–30	◆ Bisphenol A diglycidyl ether-bisphenol A copolymer	10–20
	◆ Phenolic epoxy resin	1–5	◆ Epoxy resin	1–6
	◆ Strontium chromate (carcinogen)	1–5	◆ 2-Propoxyethanol	1–5
	◆ Phenolic resin	1–5	◆ Aromatic amide curative	1–5
	◆ Epoxy/phenolic resin	1–5	◆ Bisphenol A-epichlorohydrin-formaldehyde copolymer	1–5
	◆ Methanol	0.1–1	◆ Isopropyl alcohol	1–5
			◆ Triphosphoric acid, aluminum salt (1:1)	1–5
			◆ Acetone	0.5–1.5
			◆ Zinc phosphate	<1
			◆ Oxirane, mono[(C12-14-Alkyloxy)methyl]derivatives	<0.75
			◆ Zinc Oxide	<0.1
Volatile Organic Compounds [g/L]	780–800		80–84	
Density [g/cm^3^]	0.875		1.04–1.09	
Specific Gravity [g/cm^3^]	0.88		1.06	
Flash Point [°F]	29		108.5	
Boiling Point [°F]	176		212	
Flammable Limits [% by Vol]	LEL: 1.8; UEL: 10.0		LEL: 1.5; UEL: 12.7	
Vapor Pressure [mmHg]	86		15	
Solubility in Water	Slight		Complete	

**Table 4 materials-15-08733-t004:** Primer application technologies.

Technology	Method of Applying BR-127 or EW-5000 AS Primer
Tech. 1	Hardening of the primer in accordance with the recommendations of the 3M manufacturer [22,23]
Tech. 2	Hardening of the primer and the adhesive film in one operation
Tech. 3	Primer curing in room conditions

**Table 5 materials-15-08733-t005:** Research program.

Primer	Kind of Technology	Seasoning Method	Number of Samples		
			Wedge Test	Tensile Shear Test	Corrosion Test
Samples without primer	X	X	X	X	4
BR 127	Tech. 1	Laboratory conditions	2	4	4
		Tropical conditions	3	5	
		Water conditions	3	5	
	Tech. 2	Laboratory conditions	2	4	
		Tropical conditions	3	5	
		Water conditions	3	5	
	Tech. 3	Laboratory conditions	2	4	
		Tropical conditions	3	5	
		Water conditions	3	5	
EW 5000 AS	Tech. 1	Laboratory conditions	2	4	4
		Tropical conditions	3	5	
		Water conditions	3	5	
	Tech. 2	Laboratory conditions	2	4	
		Tropical conditions	3	5	
		Water conditions	3	5	
	Tech. 2	Laboratory conditions	2	4	
		Tropical conditions	3	5	
		Water conditions	3	5	

**Table 6 materials-15-08733-t006:** Effectiveness and durability results of adhesive joints (damage characteristic of the joint after the wedge test).

Primer Application Technology and Sample Seasoning Conditions	EW-5000 AS	BR 127
TECHNOLOGY 1	Image and nature of the destruction	Image and nature of the destruction
Seasoned in laboratory conditions	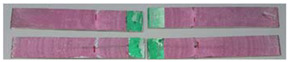 Cohesive failure	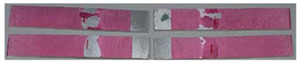 The initial cohesive character of the crack turned into adhesive during the test, constituting about 30% of the total surface
Seasoned in tropical conditions	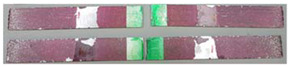 The adhesive nature of the destruction occurs on about 30% of the tested sample surface	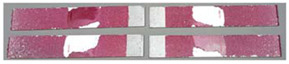 During the test, the cohesive character of the destruction was transformed into the adhesive one, occurring on 50% of the surface
Seasoned in water	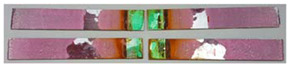 Adhesive nature of the destruction	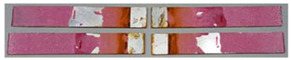 During the test, the cohesive character of the destruction was transformed into the adhesive one, occurring on 50% of the surface
TECHNOLOGY 2		
Seasoned in laboratory conditions	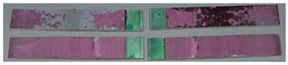 The initial, cohesive character of the crack during the test turned into adhesive, which occurs in about 60% of the entire joint	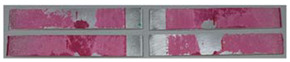 The first phase of the crack is about 60% and has a cohesive character, then turns into an adhesive character
Seasoned in tropical conditions	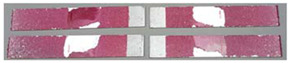 Adhesive character, which accounts for approximately 60% of the total joint	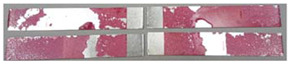 The samples show an adhesive cracking character in the area of about 60%
Seasoned in water	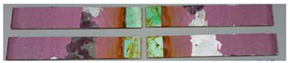 The adhesive–cohesive character of failure occurs in the ratio of 50: 50%	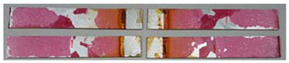 The samples show an adhesive character of cracking in the area of about 40% of the surface
TECHNOLOGY 3		
Seasoned in laboratory conditions	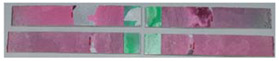 A cohesive crack in the first stage of the test turns into an adhesive failure	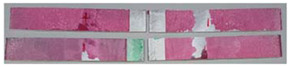 The samples are characterized by a 70% cohesive crack, which changed into adhesive character during the test
Seasoned in tropical conditions	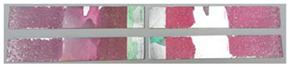 The adhesive–cohesive character of failure occurring in the ratio of 50: 50%	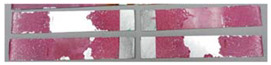 Adhesive character of destruction on the surface of about 60%
Seasoned in water	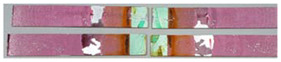 Adhesive character of destruction on the surface of about 30%	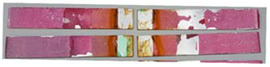 The adhesive–cohesive character of failure occurring in the ratio of 50: 50%

## Data Availability

Not applicable.

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
