# Peer review of "Durability and Corrosion Resistance Test of Adhesive Joints Using Two Adherence Promoters for the Connection of Aerospace Aluminum Alloys"

_materials, 2022, doi:10.3390/ma15248733_

Round 1

Reviewer 1 Report

please see attached.

Author Response

Thank you for all the comments and suggestions that significantly contributed to the clarity and level of the article. Most comments have been taken on board and corrections have been made to the text. The literature review was supplemented and clarified, as well as the transparency of the obtained research results. The part of the work concerning the analysis of the results was rebuilt, the work was supplemented with key information on the tested materials and the final conclusions were clarified. We hope that this huge amount of work will contribute to a positive assessment of the prepared manuscript.

Reviewer 2 Report

This manuscript performed experimental research to compare two anti-corrosion primers with different chemical compositions used in joining aerospace aluminum alloys in terms of mechanical properties and the level of anti-corrosion protection. The mechanical properties of the prepared samples were tested using two test methods: wedge tests and shear strength tests. In both cases, the samples were aged in laboratory conditions, in tap water, and in a climatic chamber (with increased temperature and humidity). The results are interesting and abundant which have indicative value for the selection of technology of making an adhesive joint. However, some discussion of experimental results are not consistent with the corresponding figures, which is kind of confusing. This manuscript needs to be mainly revised before accepted.

(1) The Introduction and literature review do not track the up-to-date research enough. More related literatures in recent five years should be added. 

(2) Section 2: (a) The resolution of Figures 1-3 is low. Please improve the quality of Figures 1-3. (b) On Page 3, the last paragraph mentions “205 x 152 mm”, however the geometrical size annotated in Figures 1 is 152.4. Please confirm which one is the correct size.

(3) Section 3: Some discussion of the results are not consistent with the corresponding figures, which is kind of confusing. Please have a careful check. (a) In Section 3.1, on Page 13, “The lowest crack propagation in the wedge test under laboratory conditions was obtained for samples produced using technology 2, which using the EW-5000 AS primer.” “The highest results were obtained for technology 1.” However, the lowest line in Figure 7 is marked with blue hollow circle which represents EW-5000 AS Tech. 1. There are similar questions for the discussion on results of samples prepared using the BR 127 primer under laboratory conditions, and for the discussion on results of the samples seasoned in water with the EW-5000 AS primer and BR 127 primer. (b) In Section 3.2, there are the similar questions with Section 3.1, such as “The lowest shear strength values were obtained for samples conditioned in laboratory conditions, prepared with the use of the EW-5000 AS primer for joints produced using technology 2.” However, the shear strength values for samples with EW-5000 AS primer laboratory conditions are Tech. 1 in Figure 10. There are several other inconsistency with the figure in this section. Please have a check. (c) It would be better if revising the title “Corrosion test results” as “3.3. Corrosion test results”.

Author Response

Thank you very much for all the comments and suggestions that significantly contributed to the clarity and level of the article. Most comments have been taken on board and corrections have been made to the text. The literature review was supplemented and clarified, as well as the transparency of the obtained research results. The part of the work concerning the analysis of the results was rebuilt, the work was supplemented with key information on the tested materials and the final conclusions were clarified. We hope that this huge amount of work will contribute to a positive assessment of the prepared manuscript.

Reviewer 3 Report

This manuscript presents a comprehensive investigation on the use of two adhesion promoters for the realization of adhesively bonded joints in aeronautical aluminum alloys. The content should be modified at several key points before it can be considered ready for publication.

1.     Abbreviations should not be used in the title. It is sufficient to indicate that two adhesion promoters are to be used, without naming them. The names of both will be specified later in the main text. However, the title only talks about mechanical properties when durability and corrosion resistance test results are given. This type of results should be mentioned in the title.

2.     The codes for both adhesion promoters (BR-127 and EW-5000-AS) appear from the beginning of the manuscript, but the manufacturer's references for each are not given until page 3, and no manufacturer's technical data sheets are referenced until then. This information should be provided to the reader much earlier in the text, as the names have no meaning without knowing what type of adhesion promoters are being discussed.

3.     Another important consideration is that at no point is the chemical composition of the adhesion promoters studied given, beyond the fact that one has chromium compounds and one does not.

4.     On page 2 it says: "The applied primers are used to... cooperate with the sol-gel." We should say that they somehow participate in the surface conversion process by means of a sol-gel reaction, and then explain in more detail what this process consists of.

5.     The discussion, on page 3, about aluminum alloys of aeronautical interest is very poor and confusing. The Al/Cu alloys mentioned in the manuscript are referred to as dural, duraluminum, duraluminium, duralum, or duralium. But it is not usual to give the names given by the authors: "duralas", "duralumin". The presentation of the Al alloys used in the presented research should be improved, giving their complete chemical composition and heat treatment state. These factors can be determinants in the interpretation of the results obtained.

6.     Reference [33] appears before references [27-32]. References should be numbered consecutively as they are mentioned in the main text.

7.     In Section "2.1. Materials for research”, it is stated that "The following 3M products..." are used. However, the adhesion promoter BR127 does not appear to be a 3M product, but a Cytec product. In addition, there is no mention at all of the aluminum alloy, when this is a key material to understand the results of the research. More detailed information on chemical compositions and properties of all the materials used in the preparation of the test samples should be given at this point, as well as references to all the manufacturers' data sheets.

8.     In Section "2.2. Preparation of the specimens" two ASTM standards are mentioned, but their references are not given. Subsequently, they are mentioned, but since they appear for the first time on page 3, it is there where their references should appear.

9.     Reference numbers should not be placed at the foot of figures (e.g., Figures 1 and 2) when such references have already been given in the main text.

10.  Referring to primer application technology as T3 (in Table 1) is confusing, because the aluminum alloys used appear to be in the T3 heat treatment condition (which is a standard designation for the state of treatment of aluminum alloys). Change the primer names of the application technologies.

11.  In general, the quality of the figures is very low and they should have the resolution indicated in the instructions for authors of the Journal.

12.  The texts in Figure 4 are not appropriate and make it difficult to understand. It is sufficient to put a letter or number next to the samples shown in the figure and give the explanation in the text at the bottom of the figure.

13.  The section "2.3. Research methodology" should be completely rewritten. The combination of types of adhesion promoters, types of application technologies, aging conditions, and types of tests performed is very confusing. There is a need for a table where all this information is collected in a comprehensive way. Additionally, the number of samples used in each test should be clearly specified in order to have a better idea of the statistical significance of the results obtained.

14.  It should be made clear that the wedge test is a durability test and is not intended to measure the mechanical strength of adhesive bonds. The designation of the type of failure (cohesive, adhesive, or mixed) and the place where it occurs (between the substrate and the primer or between the primer and the adhesive) must be clarified somewhere and a code given to each type of failure, so that it can then be accurately referred to when specifying the results of each test specimen.

15.  Figure 6 is not necessary at all, and nothing is visible from the test specimens (which, moreover, are shown inside plastic bags). This figure should be deleted.

16.  The section “3. Results and discussion” should be divided into two parts: a subsection where the results of the tests are shown, without any interpretation, and another subsection where the authors discuss the results, giving the interpretations and explanations they consider appropriate. With theway things are presented here, with results and opinions intermingled, it is difficult to see where the information ends and the interpretation begins.

17.  Tables 2, 3 and 4 should show not only the average values of crack lengths, but also the number of samples tested in each and the standard deviations of the average values obtained. The statistical significance of the results is not clear from the average values alone.

18.  Table 5 is very confusing. All the images that appear there should be deleted and replaced by only a code with the type of failure that has occurred in each case (see note 14). You can give, outside the table, some images that serve as examples of each type of adhesive bond failure, but do not put all the images inside the table itself. The printing space in a scientific journal is always scarce and valuable and should not be occupied by unnecessary content.

19.  Figures 7, 8 and 9 should be modified so that the curves are not squeezed at the top of the graphs (change the limits and scale of the vertical axis so that the curves are centered on the graphs). Graygrid lines should be removed to improve the clarity of the plots. The legend should be made more readable.

20.  In this type of test, the dispersion of results is usually large. It is therefore necessary to give error bars at each point showing the standard deviation of the average values obtained (see Note 17). This is the only way to appreciate whether the results are statistically significant or whether the differences between different types of test conditions are only due to the usual dispersion of these tests.

21.  On page 13, following Figure 9, there are several paragraphs that simply describe what the reader can see for himself in the graphs. Only the most relevant results should be highlighted, and emphasis should be placed on what differentiates one test condition from another in terms of the results obtained. For example, stating that the best results are obtained when following the adhesion promoter application recommendations given by the manufacturer does not seem very relevant. It is only when better results are obtained with a different application technology that the results are of interest, provided that an explanation can be given later as to why this is the case. These paragraphs should be completely rewritten.

22.  On page 15 there is a paragraph that specifically states: “The graphs obtained in the shear strength test do not show the influence of technology, primer, or seasoning conditions on the stiffness of the joint, with the increase in elongation being equal to the increase in force.” What seems to be clear from this comment is that the tests performed are not adequate to detect differences between the different samples, so it is useless to have performed them. Other types of tests should have been selected. Some representative graphs of the tensile shear strength tests performed should be shownand a clear explanation by the authors should be provided in the text.

23.  Figures 11-16 are redundant. One of them is sufficient to show their appearance after accelerated aging tests under the various conditions used.

24.  In a section showing results and discussion by the authors (page 18) it is unnecessary to refer to tests performed by others and for other types of issues (thermography, acoustic impedance, laser shearography) included in the references [38, 39]. This information, if considered relevant, should be placed in the Introduction and not at the end of the manuscript.

25.  In the Conclusions, the importance of the chemical composition of the primers is mentioned, butthese compositions do not appear in this paper. So, it cannot be placed as a conclusion of the research carried out.

26.  In Conclusion 1, it is reiterated that the wedge test is used to determine the strength of the adhesive bonds. This is a serious flaw in the argument presented. This test only gives an estimate of the durability of the bonds, not their strength. The substitution of one primer for another, as stated in the conclusion, cannot be based on this type of test. Only the tensile shear strength test gives, in fact, information on the strength of the joints. But the wedge test does not.

27.  Conclusion 2 states that the strength properties are higher for the EW-5000 AS primer type.However, again, it is stated that the wedge test provides strength results. This is not correct. Only the tensile shear strength test provides such information. However, not a single graph of the force-displacement curve of these tests is presented to prove it, and further stated (see note 22) that no differences are seen between them in terms of application technology, primer, or seasoning conditions.

28.  Conclusion 3 states that the presence of moisture reduces the mechanical strength of the adhesive bonds. This is a well known and widely accepted fact; it is not necessary to put references to support it, and even less in the chapter of conclusions. In fact, it is not necessary to do any kind of research to confirm that the strength is reduced, unless quantitative data is given on how much the strength value is reduced for each specimen tested. It is these percentages of reduction in mechanical strength that should appear in the conclusions.

29.  Conclusion 4 is difficult to understand. It simply gives an isolated numerical value, but does not present in a general way the results obtained in the corrosion tests. The whole section “4. Conclusions” should be rewritten so that it is not a repetition of the results presented above, but shows the conclusions drawn from the results obtained in the investigation. Conclusions cannot be a mere summary of results.

30.  Some of the DOI links in the references do not work properly (e.g., [16]). It should be verified that all of them are correctly written and updated, and those that do not appear should be completed.

31.  A complete proofreading of the text in English by a native English speaker must be performed. Some expressions are difficult to understand, and some of the vocabulary is inappropriate.

Author Response

(The authors gave the same response as above.)

Round 2

Reviewer 1 Report

The authors have essentially all of my queries.   I recommend quantifying strain energy release rate for the measurements, though it is okay if the authors decide to ignore it. 

Reviewer 2 Report

There is no comments and this article can be published.